# Current Status and Correlation of Physical Activity and Tendency to Problematic Mobile Phone Use in College Students

**DOI:** 10.3390/ijerph192315849

**Published:** 2022-11-28

**Authors:** Wen-Xia Tong, Bo Li, Shan-Shan Han, Ya-Hui Han, Shu-Qiao Meng, Qiang Guo, You-Zhi Ke, Jun-Yong Zhang, Zhong-Lei Cui, Yu-Peng Ye, Yao Zhang, Hua-Lan Li, He Sun, Zhan-Zheng Xu

**Affiliations:** 1Physical Education College, Yangzhou University, Yangzhou 225127, China; 2Institute of Sports Science, Nantong University, Nantong 226019, China; 3Institute of Sports Science, Kyunggi University, Suwon 449701, Republic of Korea; 4Physical Education College, Shangqiu University, Shangqiu 476000, China; 5School of Physical Education, Shanghai University of Sport, Shanghai 200438, China; 6School of Physical Education, Henan University of Economics and Law, Zhengzhou 450046, China; 7Physical Education College, Shangqiu Normal University, Shangqiu 476000, China; 8School of Physical Education, Jing-Gang-Shan University, Ji’an 343009, China; 9Institute of Sports and Health, Zhengzhou Shengda University, Zhengzhou 451191, China; 10School of Physical Education, Jiangxi University of Applied Science, Nanchang 330100, China; 11School of Physical Education, Heihe University, Heihe 164300, China; 12School of Physical Education, Zhengzhou University (Headquarters), Zhengzhou 450001, China

**Keywords:** physical activity, problematic mobile phone use, addictive behavior, college students, health promotion

## Abstract

Objective: To explore the effect of problematic mobile phone use on college students’ physical activity and their relationships. Methods: A cross-sectional study was conducted among 3980 college students from three universities in Jiangsu province by random cluster sampling. The International Physical Activity Questionnaire Short (IPAQ-SF) measured college students’ physical activity. The Mobile Phone Addiction Tendency Scale for College Students (MPATS) measured problematic mobile phone use tendencies. College students’ physical activity was measured by the International Physical Activity Questionnaire Short (IPAQ-SF), and the Mobile Phone Addiction Tendency Scale measured their mobile phone addiction tendency for College Students (MPATS). Results: (1) The proportions of the low-, medium-, and high-intensity physical activity were 83.5%, 10.7%, and 5.8%, respectively, with gender differences; The score of problematic mobile phone use tendency was 38.725 ± 15.139. (2) There were significant differences in problematic mobile phone use tendency among college students with different physical activity intensity (F = 11.839, *p* < 0.001, η^2^ = 0.007). (3) The level of physical activity was significantly correlated with the tendency of problematic mobile phone use (*r* = −0.173, *p* < 0.001). (4) Physical activity of college students could significantly predict the tendency of problematic mobile phone use (F (3,3605) = 11.296, *p* < 0.001). Conclusions: The physical activity of college students was mainly moderate to low intensity, while the tendency of problematic mobile phone use was high. College students’ physical activity level was one of the important constraints of problematic mobile phone use tendency.

## 1. Introduction

Problematic mobile phone use primarily refers to addiction to behavior or technology addiction caused by individuals who cannot control their use of mobile phones and other electronic products [1,2,3]. Research into problematic mobile phone use began in 2001, and the construct was developed by YouGov PLC, a research organization in the UK. The alternate usage with mobile phone dependence also includes “mobile phone addiction” and “mobile phone dependence”. These terms may have different expressions, but they all refer to the unreasonable and inappropriate use of mobile phones. The phenomenon of problematic mobile phone use characterized by excessive use has attracted the attention of psychologists, educators, physicians, sociologists, and other disciplines [4,5,6]. Symptoms of phone addiction include restlessness and anger when the Internet is disconnected, anxiety and worry when the battery runs out, and constant reminders of higher functional and behavioral disorders related to the phone’s Internet speed and configuration requirements [4]. With the development of mobile Internet technology, mobile phones play an important role in every aspect of college students’ life, including course learning, shopping consumption, transaction payment, and social communication [5,6]. Studies have shown that during the COVID-19 pandemic, problematic mobile phone use has intensified in various populations in China [7,8,9]. At the same time, the negative impact due to problematic mobile phone use is also obvious [10,11]. Exploring possible limiting factors for problematic mobile phone use is essential in this context.

The addiction is closely related to individual psychological aspects [12,13]. The research on phone addiction has so far focused on patterns and motivations of phone use [14], demographic indicators [15], individual emotional experience [16], family, social, and environmental factors [17], and self-control, subjective well-being [13] and other psychological factors. Some studies have begun investigating the relationship between mobile phone dependence and physical activity in students in recent years, such as the internal relationship between physical activity measures such as step count and calorie consumption and problematic mobile phone use [18]. Furthermore, some studies have linked problematic mobile phone use with sedentary behavior and cardiopulmonary fitness and further explored the effects of problematic mobile phone use on health and physical fitness [19]. Scholars at home and abroad have reported a relationship between problematic mobile phone use and physical activity. However, research on this topic is scarce from the perspective of physical activity among college students.

Physical activity increases energy expenditure caused by skeletal muscle contraction [20]. Insufficient physical activity linked to a sedentary lifestyle is one of the four leading risk factors for death from noncommunicable diseases [20]. Exercise and health promotion research has recently developed rapidly, particularly in physical and mental health promotion based on physical activity behavior change. It has been shown that even accumulating amounts of less than 10 min of physical activity can help improve mental and physical health. Several studies have shown that any period of moderate to vigorous physical activity can contribute to the benefits of cumulative physical activity, both physically and mentally [20,21]. With the recent decline in physical activity among different groups caused by the global coronavirus pandemic [22,23,24,25], physical activity-related health problems related to physical activity are becoming more prevalent.

Sports medicine is an effective method of promoting health and preventing disease. Is there an inherent link between physical activity and problematic mobile phone use? It is still unclear whether physical activity can alleviate or reduce the symptoms of problematic mobile phone use in college students. In other words, if there is some association between physical activity and problematic mobile phone use, whether the probability of problematic mobile phone use can be reduced by means of exercise intervention will be a very meaningful topic. Therefore, this study explores the relationship between physical activity and problematic mobile phone use using empirical research paradigms and provides a reference to slow the deepening symptoms of problematic mobile phone use by increasing physical activity among college students

## 2. Method

### 2.1. Participants

Surveys were conducted among college students using random cluster sampling. Students were from three universities in Yangzhou University, Yangzhou Vocational and Technical College in Yangzhou City, Jiangsu Province, and Nantong University in Nantong City in September 2022. The minimum sample size was calculated by Equation (1) [26]. The type I error *α* was set as 0.05, the allowable error *δ* at 0.01, and the sample rate *ρ* was set as 0.05. According to the official university websites, the total number of undergraduates in the three universities was 104,731 (data updated in 2021). The minimum sample size required for this study was *n* = 1794
(1)$$n=\frac{{\left(\frac{Z\alpha }{\delta }\right)}_{}^{2}*p*(1-p)}{1+[{\left(\frac{Z\alpha }{\delta }\right)}_{}^{2}*p*(1-p)]/N}$$

A total of 3980 questionnaires were distributed using the Questionnaire Star software 1.0 according to the administrative classes. After excluding the questionnaires with missing key information such as age and other variables, 3609 valid questionnaires were collected, with an effective rate of 90.68%. The sample size met the minimum sample size requirement. The distribution of study subjects is shown in Table 1.

### 2.2. Measurement

#### 2.2.1. Physical Activity

The Physical Activity Questionnaire using the Short Volume International Physical Activity Questionnaire (IPAQ-SF), IPAQ-SF was developed by the International Working Group on Physical Activity Measurement in 2001. The IPAQ-SF consists of seven questions, six asking about the physical activity of different intensities and the seventh asking individuals about their sedentary time. The cumulative daily time and weekly frequency of participation in low-intensity physical activity (LPA), moderate-intensity physical activity (MPA), and vigorous physical activity (VPA) were recorded over seven days. Relevant studies show that IPAQ-SF has high reliability and validity in Chinese college students [27,28].

Physical activity energy expenditure in this study was calculated as follows [29]: the assignment value of exercise intensity corresponding to a specific physical activity × weekly frequency (d/w) × time per day (min/d), and the energy expenditure of the three kinds of intensity were added up to the total physical activity expenditure of a week. The MET of VPA was 8.0, MPA was 4.0, and walking was 3.3. LPA was defined as meeting either of the following criteria: (1) no reported physical activity and (2) reported physical activity that did not meet the following high or moderate grouping criteria. Any of the following three inclusion criteria must have been met: (1) the total number of days of high-intensity physical activity of all kinds should be at least three and the daily activity time should be at least 20 min; (2) all kinds of moderate-intensity or/and walking activities total ≥ 5 days, and daily activity time ≥ 20 min; (3) the total physical activity of the three intensities should be ≥5 days, and the total physical activity should be ≥ 600 MET-min/w per week. VPA was defined by meeting either of the following criteria: (1) the total number of high-intensity physical activity days of all types for at least three days and physical activity for at least 1500 MET-min/w per week; (2) the total number of physical activity days of the three intensities should have been ≥7 days, and the weekly physical activity for ≥ 3000 MET min/w.

#### 2.2.2. Problematic Mobile Phone Use

The Mobile Phone Addiction Tendency Scale (MPATS) was developed by Chinese scholars Xing [30]. This scale had 16 items with four dimensions. The 5-point Likert scale ranged from 1 to 5 for completely disagree, not quite disagree, relatively agree, and completely agree. The scale scores range from 16 to 80, with higher scores indicating more serious phone addiction tendency. The load of four factors on the scale ranged from 0.51 to 0.79, and the cumulative variance contribution rate was 54.3%. The results of the confirmatory factor analysis showed that the four-factor model fit well. The Cronbach α coefficient of the total scale was 0.83, and the α coefficients of the four factors ranged from 0.55 to 0.80. The test-retest reliability of the total scale was 0.91, and the test-retest reliability of the four factors ranged from 0.75 to 0.85. MPATS is generally suitable for measuring problematic mobile phone use in Chinese college students.

### 2.3. Data Analysis

SPSS 25.0 and EXCEL software 2016 were used for data processing (1). Firstly, the were preprocessed and retested while eliminating the missing data; (2) Descriptive data were used to analyze the status quo of college students’ physical activity level and problematic mobile phone use tendency. The Chi-square test was used to analyze the differences in physical activity levels between genders and grades (Cramer’s V coefficient was used as the effect size). One-way analysis of variance (ANOVA) was used to analyze the differences in problematic mobile phone use tendency among students of different genders and grades with the effect size of η^2^. According to Cohen’s criterion, the effect size in ANOVA was measured by η^2^, and the η^2^ values of the small, medium and large effect sizes were agreed to correspond to 0.01, 0.06, and 0.14, respectively [31]. (3) One-way analysis of variance was used to analyze the differences in problematic mobile phone use tendencies among college students under different physical activity intensities. (4) Kendall’s correlation analysis was used to examine the correlation between college students’ problematic mobile phone use tendencies (including withdrawal symptoms, highlight behavior, social comfort, and mood change) and physical activity level index. (5) Linear hierarchical regression analysis was used to verify the predictive effect of college students’ physical activity level on problematic mobile phone use tendency and its dimensions. The variables were standardized (Z-score) before adjusting the effect.

## 3. Results

### 3.1. Descriptive Analysis

Table 2 shows that the physical activity status of Chinese college students in low, medium, and high intensity accounted for 83.5%, 10.7% and 5.8% of the total number, respectively, and the physical activity status of low intensity accounted for the majority. Regarding gender, the low-, medium-, and high-intensity physical activity of boys accounted for 75.5%, 14.8%, and 9.8%, respectively, while the low-, medium-, and high-intensity physical activity accounted for 92.4%, 6.2%, and 1.3%, respectively. The LPA level of girls was higher than that of boys, and the MVPA level of boys was higher than that of girls. There were significant differences between male and female students in the status of low, medium, and high physical activity status (*p* < 0.001, Cramer’s V = 0.238). Furthermore, from the grade perspective, the LPA level of freshman students accounted for the most significant proportion (86.1%). The VPA level of senior students accounted for the most considerable proportion (7%); however, overall, there was no difference in college students’ physical activity by grade (*p* = 0.06).

Table 3 shows that in the analysis of Chinese college students’ problematic mobile phone use and gender and grade differences, the two dimensions of withdrawal symptoms and social comfort were found to have no significant differences in gender and grade (*p* > 0.05), and the scores show a relatively average trend. In the statistical analysis of highlight behavior and gender, boys scored higher than girls on average, and there was a significant difference between male and female students (*p* < 0.001), and in the dimension of highlight behavior, students of different grades also showed a significant difference (*p* < 0.001). From the dimension of mood change analysis, Chinese college students showed significant differences in gender and grade (*p* < 0.01), and the score of male students was generally higher than that of female students, showing a strong tendency for problematic mobile phone use.

### 3.2. Analysis of Problematic Mobile Phone Use Tendency to Mobile Phones of College Students with Different Intensities of Physical Activity

Table 4 shows that according to the four aspects of withdrawal symptoms, highlighting behavior, social comfort, and mood alteration under the different intensities of physical activity. There were significant differences between different intensities of physical activities and problematic mobile phone use tendencies (*p* < 0.001, η^2^ = 0.007). The score of problematic mobile phone use under LPA was significantly higher than that of MVPA. In the dimension of withdrawal symptoms, the scores of addictions among different levels of physical activity were statistically significant (*p* < 0.001, η^2^ = 0.006), and the scores of problematic mobile phone use scores under LPA were significantly higher than those of MVPA. In the dimension of highlight behavior, there were significant differences in addiction scores between different levels of physical activity (*p* < 0.01, η^2^ = 0.003). In the dimension of social comfort, there were significant differences in addiction scores among different levels of physical activity (*p* < 0.001, η^2^ = 0.011). In the dimension of mood alteration, there were significant differences in the addiction scores between different intensities of physical activity (*p* < 0.01, η^2^ = 0.005), and the scores of problematic mobile phone use under LPA were significantly higher than those of MVPA. The lower the physical activity rating, the higher the propensity to become addicted to mobile phones.

### 3.3. Correlation Analysis

Table 5 shows the correlation between the level of physical activity and the tendency to addiction to mobile phones, and its dimensions ranges from −0.151 to −0.193. The correlation between the total score of problematic mobile phone use tendency and each dimension ranged from 0.737 to 0.848, and the correlation between each dimension ranged from 0.643 to 0.715.

### 3.4. Regression Analysis

To further explore the internal relationship between physical activity and mobile phone dependence, based on ANOVA and correlation analysis of physical activity level on problematic mobile phone use tendency and its dimensions, a multiple hierarchical regression analysis was used to investigate the predictive effect of physical activity level on problematic mobile phone use tendency and its dimensions. The outcome variables were used as the total score of addiction tendency, withdrawal symptoms, social comfort, and mood change. The predictors of gender and grade were placed in the first layer as Model 1. Physical activity levels were placed in layer 2, which served as Model 2.

Table 6 shows that for the total score of problematic mobile phone use, the predictor variable of physical activity (F (3,3605) = 11.296, *p* < 0.001) had a significant predictive effect on the regression model, which could explain 0.9% of the total variability of problematic mobile phone use. Demographic indicators (F (2,3606) = 1.132, *p* = 0.322) had no significant predictive effect on the regression model, and gender (β = −0.05, *p* < 0.01) was a significant predictor. It has a negative predictive effect. In the dimension of withdrawal symptoms, the predictor variable of physical activity (F (3,3605) = 7.651, *p* < 0.001) had a general predictive effect on the regression model, which could explain 0.6% of the variability of truncated symptoms, while the demographic index (F (2,3606) = 0.787, *p* < 0.001) had no significant predictive effect on the regression model. In the dimension of highlight behavior, the predictors demographic sociological index (F (2,3606) = 21.764, *p* < 0.001) and college student physical activity (F (3,3605) = 22.511, *p* < 0.001) were both significant predictors of the regression model and could explain 0.1% and 0.6% of the variability of highlight behavior, respectively. Gender (β = −0.1, *p* < 0.001) and college students’ physical activity (β = 0.079, *p* < 0.001) were significant predictors. College students’ physical activity was positively correlated with gender, while gender had a negative correlation. In the dimension of social comfort, the predictor variable of physical activity (F (3,3605) = 19.331, *p* < 0.001) had a significant predictive effect on the regression model, which could explain 1.6% of the variability of social comfort, while the demographic index (F (2,3606) = 2.646, *p* > 0.05) had no significant predictive effect on the regression model, where gender (β = −0.34, *p* < 0.05) was a significant predictor variable and had a negative predictive effect. In the dimension of mood alteration, the predictor variable college student physical activity (F (3,3605) = 8.053, *p* < 0.001) had a significant predictive effect on the regression model, which could explain 0.7% of the variability of mood change, while the demographic index (F (2,3606) = 1.123, *p* > 0.05) had no significant predictive effect on the regression model; gender (β = −0.045, *p* < 0.001) was a significant predictor variable and had a negative predictive effect.

The R^2^ value in multiple regression analysis determines the effect size. In the analysis of this study, we graded effect sizes using Cohen criteria. According to Cohen’s standard, R^2^ values of 0.02, 0.13, and 0.26 are small, medium, and large effects, respectively [31]. As evident from the regression results in Table 6, demographic indicators have small effects on the variation explanation of the total score of problematic mobile phone use (R^2^ = 0.007), withdrawal symptoms (R^2^ = 0.001), social comfort (R^2^ = 0.001) and mood alteration (R^2^ = 0.001). The explanation of variation in highlight behavior (R^2^ = 0.012) was close to the standard of small effect; the physical activity level of college students had a small effect on the variation explanation of the total score of problematic mobile phone use (R^2^ = 0.009), withdrawal symptoms (R^2^ = 0.006), and mood alteration (R^2^ = 0.007). In contrast, the variation explanation of highlighting behavior (R^2^ = 0.018) and social comfort (R^2^ = 0.016) was close to the standard of small effect.

## 4. Discussion

In the mobile Internet era, mobile phones play an increasingly important role in college students’ studies and life. In this study of Yangzhou city, Jiangsu province, three university student’s physical activity, and cell phone addiction conditions were investigated.

The current college students’ physical activity level and the inner link between problematic mobile phone use was discussed in ordinary colleges and universities through reasonable and effective physical activity to prevent and improve problematic mobile phone use and to further explore the mechanism of physical activity and problematic mobile phone use in different populations.

The physical activity level of college students was low, which is consistent with the survey results of Yang et al. [32] on the physical exercise of college students in Guangzhou. Exercise intensity, duration, and amount were low among college students in Guangzhou, which may reduce the positive effects of exercise on problematic mobile phone use. Both Xia et al. and Wilson et al. found that college students engaged in medium or low levels of physical activity, which is consistent with this study’s results. The level, intensity, duration, and frequency of physical activity of male college students were higher than those of female students, which was consistent with the results of previous studies [33]. It is well known that sports are often associated with fighting and winning, reflecting social expectations of male personality traits. Boys are more physically active than girls, while stereotypes label women quiet and docile.

College students scored higher in the problematic mobile phone use tendency. College students had the highest score on the dimension of withdrawal symptoms, indicating a strong desire to own or use mobile phones and significant physical and psychological reactions to being without mobile phones. In the context of the epidemic, this is consistent with the findings of Tian Yu et al. [34]. Regular online classes, lockdowns, and home isolation during the COVID-19 pandemic may be one of the significant reasons for college students’ persistently high levels of negative emotions and mobile phone dependence [35]. Male college students scored higher than female college students in the total score of problematic mobile phone use tendency and the dimension of highlight behavior and mood alteration. In contrast, female college students scored higher than males on the dimension of withdrawal symptoms and social comfort. These differences could be attributed to the influence of personality traits and gender cognition. College students have gender differences in attention preferences and lifestyles (for example, boys tend to use mobile games and other operational activities, while girls tend to browse media consulting or network social activities) [36]. Srividya [37] and Omari [38] noted in their research that female college students use mobile phones more frequently in interpersonal communication, entertainment, and online shopping compared to male students. These behaviors are also communicated through mobile phones. However, regarding the use of the Internet, the research of The Naming et al., maintaining and storing mobile phones in the era of mobile Internet, scientific and reasonable use of mobile phones by college students, is conducive to learning progress and campus adaptation, and reduces the possibility of mental health problems [39]. Using mobile phones moderately or excessively will result in completely different results for college students’ studies and lives.

The analysis of the differences between college students’ problematic mobile phone use tendencies under different intensities of physical activity shows that with the constant improvement of the college students’ physical activity level, problematic mobile phone use tendency, and each dimension score also are on the decline. In low to moderate intensity levels of physical activity, college students’ addictions and total scores were significantly lower. Moreover, the total score of problematic mobile phone use tendency and its dimensions under LPA was the highest, while the total score of problematic mobile phone use tendency and its dimensions under MPA was the lowest. There was a more significant positive impact of MPA on college students’ dependence on mobile phones than VPA. The positive effects of physical activity on college students’ tendency to addiction to mobile phones can only be maximized after the MPA has been reached. MPA better measured college students’ mobile phone dependence than VPA. Furthermore, LPA cannot effectively reduce problematic mobile phone use among college students. McMorris et al.’s meta-analysis of acute exercise of different intensities on cognitive speed and accuracy found that moderate-intensity exercise had a significantly more significant impact than low-intensity and high-intensity exercise in terms of speed, providing compelling evidence of this study’s findings and results [40].

The level of physical activity, cell phone addiction tendency, withdrawal symptoms, social comfort, highlighting behavior, and mood alteration in the four dimensions of Chinese college students were significantly negatively correlated. According to previous research results, for college students who engage in physical activity, the greater the intensity, the longer the duration, and the greater the frequency, the lower their likelihood of becoming addicted to mobile phones [32,41,42]. People’s primitive instincts can be satisfied by a series of characteristics of the network, including its openness, virtue, concealment, richness, convenience, and stimulation from the point of view of instinct [43]. Active exercise interventions have been developed based on instinct, implying that sports have an instinct in addition to the effects of internet activation. This leads to the conclusion that positive movement “addiction” may replace the Internet (cell phones, computers). Rather than a deliberately suppressed negative “addiction,” the displacement is derived from the channel of instinct, transfer, and exchange [41,44,45,46]. The results also showed that different intensity exercise levels had different effects, and moderate-intensity exercise was better than low-intensity and high-intensity exercise. Therefore, moderate-intensity physical activity prevented and improved phone addiction the most among the physical activity scales. Moreover, low-intensity physical activity is unlikely to affect college students’ problematic mobile phone uses positively.

The regression analysis results of this study show that college students’ physical activity can predict the tendency of problematic mobile phone use: the higher the level of physical activity, the lower the problematic mobile phone use of college students. Physical activity has been recognized by scholars worldwide. It has been accepted and recognized by the public worldwide because physical activity has the fitness function of improving and optimizing the nervous system, improving metabolism, enhancing immunity, and strengthening the physique. The process of physical activity also gives the body the feeling of power, accomplishment, and happiness that can help eliminate negative emotions such as anxiety, depression, and exercise pleasure [42,47,48]. Numerous studies have shown that [42,49,50,51] physical exercise is effective in treating mental diseases and addiction diseases. There is a close relationship between problematic mobile phone use and college students’ mental health. For example, Elahi JD et al. [52] pointed out excessive smartphone use is related to fear of information loss and depression. Yuan et al. [53] further confirmed this correlation and pointed out that online gaming disorders and excessive use and dependence on smartphones directly affected the severity of depression. Li et al. [54] also showed that problematic cellphone use may be a potential risk factor for circadian sleep disorder in Chinese college students. Sleep problems may aggravate anxiety, depression, and other psychological diseases in college students. The positive effect of physical activity on college students’ problematic mobile phone use may be derived from its function of promoting mental health, which should be further promoted and applied in the intervention of college students’ problematic mobile phone use and other psychological and behavioral problems in the future.

This study on college students’ physical activity and its relationship to problematic mobile phone use summarized students’ problematic mobile phone use prevention and intervention for the mobile Internet era and provided empirical support and beneficial revelations. Due to limited access to various aspects, especially in the new outbreak investigation background, this study may have some limitations. Firstly, cross-sectional studies based on questionnaires make it difficult to accurately determine the causal relationships between variables, let alone the long-term effects between variables. A high-quality experimental design longitudinal study will be considered to empirically test the relationship between physical activity and problematic mobile phone use. The second issue is that selecting research objects and sampling methods still requires improvement and perfection. Future research will consider broadening the scope of the survey, accurately selecting representative cities or college students as survey objects, increasing the sample size, and considering the representative sample further to validate the universality and rationality of the research conclusions.

## 5. Conclusions

Problematic mobile phone use has been the focus of research in psychology, education, medicine, and sociology, as well as in the internet addiction. The results of this study show that college students’ physical activity levels were mostly moderate to low, and that the tendency of problematic mobile phone use was high. Physical activity is one of the main constraints to problematic mobile phone use. Future research can establish a new model by introducing relevant mediating or moderating variables between the two and conducting chain multiple mediating effect tests or moderating effect tests to clarify further the occurrence and boundary mechanism of physical activity and problematic mobile phone use.

## Figures and Tables

**Table 1 ijerph-19-15849-t001:** Distribution of research objects.

		Frequency	Percentage
Gender			
	males	1891	52.4
	females	1718	47.6
Grade			
	1	1353	37.5
	2	976	27.0
	3	1050	29.1
	4	230	6.4
Total	Total	3609	100

**Table 2 ijerph-19-15849-t002:** Physical activity status and gender and grade differences among college students.

			Low	Middle	High	X^2^	*p*	Cramer’s V
Total								
		*n*	3015	386	208			
		%	83.5	10.7	5.8			
Gender								
	male					203.6	<0.001	0.238
	(*n* = 1891)	*n*	1427	279	185			
		%	75.5	14.8	9.8			
	female							
	(*n* = 1718)	*n*	1588	107	23			
		%	92.4	6.2	1.3			
Grade								
	1	*n*	1165	125	63	12.1	0.06	0.058
	(*n* = 1353)	%	86.1	9.2	4.7			
	2	*n*	804	114	58			
	(*n* = 976)	%	82.4	11.7	5.9			
	3	*n*	862	117	71			
	(*n* = 1050)	%	82.1	11.1	6.8			
	4	*n*	184	30	16			
	(*n* = 230)	%	80	13	7			

**Table 3 ijerph-19-15849-t003:** Current situation and gender and grade differences of problematic mobile phone use tendency in college students.

Aggregate Score		M	SD	F	*p*	η²
		38.725	15.139			
Gender						
	male (*n* = 1891)	39.077	15.793	2.139	0.144	0.001
	female (*n* = 1718)	38.339	14.379			
Grade						
	1 (*n* = 1353)	38.334	14.226	1.477	0.219	0.001
	2 (*n* = 976)	39.551	15.523			
	3 (*n* = 1050)	38.390	15.789			
	4 (*n* = 230)	39.052	15.605			
Withdrawal symptoms		15.516	5.925			
Gender						
	male (*n* = 1891)	15.491	6.079	1.095	0.35	0.001
	female (*n* = 1718)	15.543	5.752			
Grade						
	1 (*n* = 1353)	15.580	5.672	0.069	0.793	<0.001
	2 (*n* = 976)	15.725	6.001			
	3 (*n* = 1050)	15.278	6.145			
	4 (*n* = 230)	15.335	6.029			
Highlight behavior		8.523	4.080			
Gender						
	male (*n* = 1891)	8.831	4.276	22.68	<0.001	0.006
	female (*n* = 1718)	8.185	3.827			
Grade						
	1 (*n* = 1353)	8.038	3.789	11.44	<0.001	0.008
	2 (*n* = 976)	8.906	4.199			
	3 (*n* = 1050)	8.661	4.236			
	4 (*n* = 230)	9.130	4.224			
Social comfort		7.636	3.224			
Gender						
	male (*n* = 1891)	7.635	3.295	<0.001	0.983	<0.001
	female (*n* = 1718)	7.637	3.145			
Grade						
	1 (*n* = 1353)	7.738	3.189	2.34	0.071	0.002
	2 (*n* = 976)	7.746	3.207			
	3 (*n* = 1050)	7.444	3.285			
	4 (*n* = 230)	7.448	3.188			
Mood alteration		7.050	3.110			
Gender						
	male (*n* = 1891)	7.120	3.207	11.99	<0.001	0.008
	female (*n* = 1718)	6.973	3.000			
Grade						
	1 (*n* = 1353)	6.978	3.009	8.889	0.006	0.006
	2 (*n* = 976)	7.174	3.163			
	3 (*n* = 1050)	7.008	3.177			
	4 (*n* = 230)	7.139	3.169			

**Table 4 ijerph-19-15849-t004:** The difference analysis of the problematic mobile phone use tendency of college students under different physical activity intensity.

		Low (*n* = 3015)		Middle (*n* = 386)		High (*n* = 208)	
		M	SD	M	SD	M	SD
Totality							
	mark	39.230	14.838	35.352	15.043	37.668	18.488
	F	11.839					
	*p*	<0.001					
	η^2^	0.007					
Withdrawal symptoms							
	mark	15.703	5.815	14.256	5.898	15.135	7.162
	F	10.719					
	*p*	<0.001					
	η^2^	0.006					
Highlight behavior							
	mark	8.605	4.031	7.829	3.979	8.635	4.810
	F	6.282					
	*p*	0.002					
	η^2^	0.003					
Social comfort							
	mark	7.783	3.178	6.839	3.168	6.981	3.682
	F	19.428					
	*p*	<0.001					
	η^2^	0.011					
Mood alteration							
	mark	7.139	3.058	6.427	3.119	6.918	3.680
	F	9.183					
	*p*	<0.001					
	η^2^	0.005					

**Table 5 ijerph-19-15849-t005:** Association between physical activity level and propensity for problematic mobile phone use.

	Statistics	IPAQ Grade	MPATS Total Points	Withdrawal Symptoms	Highlight Behavior	Social Comfort	Mood Alteration
IPAQ grade	r		−0.173 **	−0.165 **	−0.151 **	−0.193 **	−0.164 **
	*p*		<0.001	<0.001	<0.001	<0.001	<0.001
MPATS total points	r	−0.173 **		0.848 **	0.776 **	0.737 **	0.806 **
	*p*	<0.001		<0.001	<0.001	<0.001	<0.001
Withdrawal symptoms	r	−0.165 **	0.848 **		0.662 **	0.643 **	0.715 **
	*p*	<0.001	<0.001		<0.001	<0.001	<0.001
Highlight behavior	r	−0.151 **	0.776 **	0.662 **		0.597 **	0.704 **
	*p*	<0.001	<0.001	<0.001		<0.001	<0.001
Social comfort	r	−0.193 **	0.737 **	0.643 **	0.597 **		0.633 **
	*p*	<0.001	<0.001	<0.001	<0.001		<0.001
Mood alteration	r	−0.164 **	0.806 **	0.715 **	0.704 **	0.633 **	
	*p*	<0.001	<0.001	<0.001	<0.001	<0.001	

** The correlation was significant at level 0.01 (two-tailed).

**Table 6 ijerph-19-15849-t006:** Hierarchical regression analysis of physical activity on the trend of problematic mobile phone use and its dimensions.

Consequent Variable	Model Category	Model Summary				Significance of Predictor Variables
		R	R^2^	F	*p*	
MPATS total points						
	1	0.083	0.007	F (2,3606) = 1.132	0.322	gender (β = −0.05, *t* = −2.928, *p* = 0.003)
	2	0.096	0.009	F (3,3605) = 11.296	<0.001	
Withdrawal symptoms						
	1	0.021	0.001	F (2,3606) = 0.787	0.455	
	2	0.08	0.006	F (3,3605) = 7.651	<0.001	
Highlight behavior						
	1	0.109	0.012	F (2,3606) = 21.764	<0.001	gender (β = −0.1, *t* = −5.859, *p* < 0.001)
	2	0.136	0.018	F (3,3605) = 22.511	<0.001	grader (β = 0.079, *t* = 4.802, *p* < 0.001)
Social comfort						
	1	0.038	0.001	F (2,3606) = 2.646	0.071	gender (β = −0.34, *t* = −1.974, *p* = 0.048)
	2	0.126	0.016	F (3,3605) = 19.331	<0.001	
Mood alteration						
	1	0.025	0.001	F (2,3606) = 1.123	0.326	gender (β = −0.045, *t* = −2.619, *p* = 0.009)
	2	0.082	0.007	F (3,3605) = 8.053	<0.001	

## Data Availability

The original contributions presented in this study are included in the article, further inquiries can be directed to the corresponding author.

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
