# Peer review of "Current Status and Correlation of Physical Activity and Tendency to Problematic Mobile Phone Use in College Students"

_ijerph, 2022, doi:10.3390/ijerph192315849_

Round 1

Reviewer 1 Report

Comments to the Author

Dear Editor and authors:

Thank you for the opportunity to review your manuscript. This study aims to examine the association between physical activity and tendency to mobile phone addiction in college students. This study explores an interesting research question, albeit with a major research limitation of being cross-sectional in design. While the paper has a number of strengths, there are a number of issues that need to be addressed before it can be considered for publication. 

General:

A revision of the English language is recommended.

Introduction

Page 2 line 48 Authors should add some reference after this sentence: “The phenomenon of mobile phone addiction characterized by excessive use has attracted the attention of psychologists, educators, physicians, sociologists, and other disciplines.” 

Page 2 line 54-56 The sentence should be modified as: With the development of mobile Internet technology, mobile phones play an important role in every aspect of college students’ life, including …….

Page 2 line 56 “Studies have shown that during the COVID-19 pandemic, mobile phone addiction has intensified in various populations in China.” Is it just China? How many other countries have similar phenomenon?

Page 2 line 64 The author reported that “some studies have begun investigating the relationship between mobile phone dependence and physical activity in college students in recent years.” “However, research on this topic is scarce from the perspective of physical activity among college students.” There is a contradiction between the two sentences.

The introduction reviews several previous studies that have answered a similar question. How then is this study novel, other than being in China? A more focused discussion on the uniqueness of this study is needed. 

Method

Page 3 line 92 change “Object” to “Participants”

Page 3 line 94 middle school students? 

Table 1 change “man/ woman student” to “Males/females”

Page 3 line 109 suggest deleting “Questionnaire” behind Physical Activity

Page 4 line 136 suggest removing “scale” behind Mobile Phone Addiction Tendency

Page 4 line 138-143 This sentence is not clear, please rewrite it. Examples can be given for each dimension.

Page 4 line 144-146 Suggest to rewrite as “the scale scores range from 16 to 80, with higher scores indicating more serious phone addiction tendency.”

Page 4 line 153 suggest deleting “Methods” behind Data Analysis

Page 4 line 163 Multivariate analysis of variance (MANOVA) should be adopt to analyze the differences in mobile phone addiction tendencies among college students under different physical activity intensities. For significant MANOVAs, follow-up one-way analysis of variance (ANOVA) and Least Significant Difference (LSD) were conducted to identify where the significant differences occurred.

Results

Page 4 line 173 change to “Descriptive analysis”

Page 4 line 176-188 Description of Table 2, there are many variables, such as low, medium, and high-intensity physical activity. There also contain LPAMVPA VPA. But we cannot find relevant variables in Table 2.

Page 6 line 194, 196, Page 8 line 250 What is salient behavior means?

Page 6 line 206-221 Consistent with the earlier comment, the author needs to reanalysis the data using Multivariate analysis of variance (MANOVA).

The author use “MPATS total points” in Table 5, and “Aggregate score” in Table 6. The description of the same variable should be unified.

Page 8 line 244-246 Demographic indicators (F (2,3606) = 1.132, p = 0.322) had no significant predictive effect on the regression model, but gender (β = -0.05, p < 0.01) was a significant predictor. I don’t know why this is happening?

While the variance in the total and four dimensions of mobile phone addiction tendency range from 0.6%-1.8%, you still feel comfortable to make ‘grand’ statements about the factor, e.g., “a significant predictive effect on the regression model, which could explain 0.6% of the variability of truncated symptoms” (Page 8 line 247-249). It would be worth considering how such statement are phrased.

Page 9 Line 267 The R2 value is a false expression. Furthermore, all the R2 are small effects (less than 0.13), What is the purpose of using a paragraph to describe it?

Discussion

Page 9 Line 288-289 The exercise time and exercise frequency are not described in the manuscripts and cannot be compared with other studies.

Page 9 Line 295-296 Same as above.

The discussion needs to be restructured to address the more novel aim –the association between physical activity and different dimensions of mobile phone addiction tendency.

Author Response

Point-by-point Responses to Reviewer 2

Dear reviewer,

Thank you for the time and effort that you have dedicated to providing your insightful and valuable comments on our manuscript. Although I do not know the situation around you, please stay healthy and keep safe. Here are point-by-point responses to your comments, I hope the responses address your concerns effectively.

Sincerely,

Comment 1:

Page 2 line 48 Authors should add some reference after this sentence: “The phenomenon of mobile phone addiction characterized by excessive use has attracted the attention of psychologists, educators, physicians, sociologists, and other disciplines.”

Response 1:

Thank you for your comment. This is a very good suggestion. We re-read the relevant papers and selected important papers for reference.

Comment 2:

Page 2 line 54-56 The sentence should be modified as: With the development of mobile Internet technology, mobile phones play an important role in every aspect of college students’ life, including …….

Response 2:

Thank you for your comment. This is also a very good suggestion. We all accept your suggestions. And we made revisions in the paper.

Comment 3:

Page 2 line 56 “Studies have shown that during the COVID-19 pandemic, mobile phone addiction has intensified in various populations in China.” Is it just China? How many other countries have similar phenomenon?

Response 3:

Thank you for your comment. Indeed, as you said, during the COVID-19 pandemic, the phenomenon of mobile phone addiction among college students has increased. Current research shows that this phenomenon occurs among university students in China, the United States, Korea, the United Kingdom and some Nordic countries[1-4]. This is also the intrinsic motivation for this study.

[1] Priya DB, Subramaniyam M. Fatigue due to smartphone use? Investigating research trends and methods for analysing fatigue caused by extensive smartphone usage: A review. Work. 2022;72(2):637-650. doi: 10.3233/WOR-205351. PMID: 35527598.

[2] Chen B, Sun J and Feng Y (2020) How Have COVID-19 Isolation Policies Affected Young People’s Mental Health? – Evidence From Chinese College Students. Front. Psychol. 11:1529. doi: 10.3389/fpsyg.2020.01529

[3] Lee, Phillip Sangwoo and Chong Min Lee. “Are South Korean College Students Benefitting from Digital Learning?” International Journal of Human–Computer Interaction (2022): n. pag.

[4] Saadeh H, Al Fayez RQ, Al Refaei A, Shewaikani N, Khawaldah H, Abu-Shanab S, Al-Hussaini M. Smartphone Use Among University Students During COVID-19 Quarantine: An Ethical Trigger. Front Public Health. 2021 Jul 26;9:600134. doi: 10.3389/fpubh.2021.600134. PMID: 34381747; PMCID: PMC8350027.

Comment 4:

Page 2 line 64 The author reported that “some studies have begun investigating the relationship between mobile phone dependence and physical activity in college students in recent years.” “However, research on this topic is scarce from the perspective of physical activity among college students.” There is a contradiction between the two sentences.

Response 4:

Thank you for your comment. In fact, what we want to say is that relevant research has been carried out in the student population, but there is less relevant research in the university student population. Thanks for your comment, we've reworked the language.

Comment 5:

The introduction reviews several previous studies that have answered a similar question. How then is this study novel, other than being in China? A more focused discussion on the uniqueness of this study is needed.

Response 5:

Thank you for your comment. This is a very good suggestion. We added more analysis to the paper to show the uniqueness of the study.

Comment 6:

Page 3 line 92 change “Object” to “Participants”.

Response 6:

Thank you for your comment. We have revised it in accordance with your comment.

Comment 7:

Page 3 line 94 middle school students?

Response 7:

Thank you for your comment. We have revised it in accordance.

Comment 8:

Table 1 change “man/ woman student” to “Males/females”

Response 8:

Thank you for your comment. This is a very good suggestion. We have revised it in accordance with your comment.

Comment 9:

Page 3 line 109 suggest deleting “Questionnaire” behind Physical Activity

Response 9:

Thank you for your comment. This is a very good suggestion. We have revised it in accordance with your comment.

Comment 10:

Page 4 line 136 suggest removing “scale” behind Mobile Phone Addiction Tendency

Response 10:

Thank you for your comment. This is a very good suggestion. We have revised it in accordance with your comment.

Comment 11:

Page 4 line 138-143 This sentence is not clear, please rewrite it. Examples can be given for each dimension.

Response 11:

Thank you for your comment. We have revised it in accordance with your comment.

Comment 12:

Page 4 line 144-146 Suggest to rewrite as “the scale scores range from 16 to 80, with higher scores indicating more serious phone addiction tendency.”

Response 12:

Thank you for your comment. We have revised it in accordance with your comment.

Comment 13:

Page 4 line 153 suggest deleting “Methods” behind Data Analysis

Response 13:

Thank you for your comment. We have revised it in accordance with your comment.

Comment 14:

Page 4 line 163 Multivariate analysis of variance (MANOVA) should be adopt to analyze the differences in mobile phone addiction tendencies among college students under different physical activity intensities. For significant MANOVAs, follow-up one-way analysis of variance (ANOVA) and Least Significant Difference (LSD) were conducted to identify where the significant differences occurred.

Response 14:

Thank you for your comment. This is a very good suggestion. We very much agree with your approach to statistical analysis. Thanks again for your comment. In fact, in the paper, our analysis ideas are consistent with yours. Your idea is to determine the strength of the association between the physical activity variable and the cell phone addiction variable directly through the MANOVAs' method. In our analysis, MANOVAs were broken down. The basic idea is this, first through one-way ANOVA to determine the difference in mobile phone addiction under different intensity physical activity. Secondly, the latent variables related to physical activity and mobile phone addiction were analyzed to determine the reasons for the differences. Finally, the linear regression method is used to determine the predictive effect between the variables. We look forward to our response to answer your questions. In addition, it should be noted that in the correlation analysis of physical activity variables, we quantified physical activity into three levels: high, medium, and low. Therefore, the analysis of physical activity variables is an analysis of hierarchy rather than an analysis of direct data.

Comment 15:

Page 4 line 173 change to “Descriptive analysis”.

Response 15:

Thank you for your comment. We have revised it in accordance with your comment.

Comment 16:

Page 4 line 176-188 Description of Table 2, there are many variables, such as low, medium, and high-intensity physical activity. There also contain LPA、MVPA VPA. But we cannot find relevant variables in Table 2.

Response 16:

Thank you for your comment. The related variables are in the header of the first row of the table. Please check it out.

Comment 17:

Page 6 line 194, 196, Page 8 line 250 What is salient behavior means?

Response 17:

Thank you for your comment. Thanks for pointing out. Spelling mistake here, it should be highlight behavior.

Comment 18:

Page 6 line 206-221 Consistent with the earlier comment, the author needs to reanalysis the data using Multivariate analysis of variance (MANOVA).

Response 18:

Thank you for your comment. This is a very good suggestion. It should be noted that in the correlation analysis of physical activity variables, we quantified physical activity as high, medium, and low. Therefore, the analysis of physical activity variables is an analysis of hierarchy rather than an analysis of direct data. Therefore, the one-way ANOVA method is suitable. We very much agree with your approach to statistical analysis. Thanks again for your comment. In fact, in the paper, our analysis ideas are consistent with yours. Your idea is to determine the strength of the association between the physical activity variable and the cell phone addiction variable directly through the MANOVAs' method. In our analysis, MANOVAs were broken down. The basic idea is this, first through one-way ANOVA to determine the difference in mobile phone addiction under different intensity physical activity. Secondly, the latent variables related to physical activity and mobile phone addiction were analyzed to determine the reasons for the differences. Finally, the linear regression method is used to determine the predictive effect between the variables. We look forward to our response to answer your questions.

Comment 19:

The author use “MPATS total points” in Table 5, and “Aggregate score” in Table 6. The description of the same variable should be unified.

Response 19:

Thank you for your comment. We have revised it in accordance with your comment.

Comment 20:

Page 8 line 244-246 Demographic indicators (F (2,3606) = 1.132, p = 0.322) had no significant predictive effect on the regression model, but gender (β = -0.05, p < 0.01) was a significant predictor. I don’t know why this is happening?

Response 20:

Thank you for your comment. This is the more critical issue. This is the case that if all demographic variables are included in the numerator in a hierarchical regression analysis, no significant effect is found. However, in independent analyses of demographic variables (mainly gender and grade), the effect of gender is more pronounced.

Comment 21:

While the variance in the total and four dimensions of mobile phone addiction tendency range from 0.6%-1.8%, you still feel comfortable to make ‘grand’ statements about the factor, e.g., “a significant predictive effect on the regression model, which could explain 0.6% of the variability of truncated symptoms” (Page 8 line 247-249). It would be worth considering how such statement are phrased.

Response 21:

Thank you for your comment. I deeply feel that you are a very serious expert. I very much agree with you. The tone words of the full text have been re-examined and revised. Thanks again for your comment.

Comment 22:

Page 9 Line 267 The R2 value is a false expression. Furthermore, all the R2 are small effects (less than 0.13), What is the purpose of using a paragraph to describe it ?

Response 22:

Thank you for your comment. We have revised it in accordance with your comment.

Comment 23:

Page 9 Line 288-289 The exercise time and exercise frequency are not described in the manuscripts and cannot be compared with other studies.

Response 23:

Thank you for your comment. We have revised it in accordance with your comment.

Comment 24:

Page 9 Line 295-296 Same as above.

Response 24:

Thank you for your comment. We have revised it in accordance with your comment.

Comment 25:

The discussion needs to be restructured to address the more novel aim –the association between physical activity and different dimensions of mobile phone addiction tendency.

Response 25:

Thank you for your comment. This is a very good suggestion. We have reorganized the discussion section to be more closely aligned with the research objectives of this study.

Reviewer 2 Report

I have read the given manuscript and found out several issues that should be corrected in order to be accepted for publishing.

1. The authors used term phone addiction in the paper. However, there are suggestions in scientific community that other term should be used instead.

``The term ``smartphone addiction`` is not an official disorder and so it is recommended to use the term problematic smartphone use. The usage of the word addiction in the context of mobile phone overuse is also problematic due to stigmatization of users.`` 

Randjelovic P, Stojiljkovic N, Radulovic N, Stojanovic N, Ilic I: Problematic Smartphone Use, Screen Time and Chronotype Correlations in University Students. Eur Addict Res 2021;27:67-74. doi.org/10.1159/000506738 

I would suggest to use the phrase problematic mobile phone use.

2. What was the power of the study? It can be calculated post hoc always. It would be useful information for readers.

3. Why 3980 as sample size since minimum number was show as 1794. Provide justification for using twice the minimum sample size.

4. Line 183: The authors  showed there are significant differences between male and female students in 3 activity groups, but we dont know where were the differences? A post hoc test would answer this question.

5. The authors used ANOVA test but not post hoc test to show the differences between specific groups. Please include post hoc test for ANOVA tables.

6. Line 355, the authors wrote that physical activity prevented and improved phone addiction. This statement would require an experimental study with treatment. In this case we have cross section design and correlation results. So correlation does not tell what is cause. The authors should change the narrative to be suited for obtain (correlation) results.

Author Response

Point-by-point Responses to Reviewer 1

Dear reviewer,

Thank you for the time and effort that you have dedicated to providing your insightful and valuable comments on our manuscript. Although I do not know the situation around you, please stay healthy and keep safe. Here are point-by-point responses to your comments, I hope the responses address your concerns effectively.

Sincerely,

Comment 1:

The authors used term phone addiction in the paper. However, there are suggestions in scientific community that other term should be used instead. ``The term ``smartphone addiction`` is not an official disorder and so it is recommended to use the term problematic smartphone use. The usage of the word addiction in the context of mobile phone overuse is also problematic due to stigmatization of users.``

Randjelovic P, Stojiljkovic N, Radulovic N, Stojanovic N, Ilic I: Problematic Smartphone Use, Screen Time and Chronotype Correlations in University Students. Eur Addict Res 2021;27:67-74. doi.org/10.1159/000506738

I would suggest to use the phrase problematic mobile phone use.

Response 1:

Thank you for your comment. This is a very good suggestion. We have revised it in accordance with your comment.

Comment 2:

What was the power of the study? It can be calculated post hoc always. It would be useful information for readers.

Response 2:

Thank you for your comment. The main motivations for this study are as follows. Due to the COVID-19 pandemic, the physical activity of college students in many parts of China has decreased significantly, so whether this decline in physical activity will cause the problem of mobile phone addiction to worsen is a question worth considering. Second, if there is a slight correlation between the two variables, then this provides a good basis for later exercise intervention. In the context of the current decline in the physical health of Chinese college students, so the Chinese government has repeatedly requested the reform of university physical education courses to increase their physical activity, our research group has done a series of related studies to try to promote the implementation of Chinese policies from the perspective of empirical research[1].

Comment 3:

Why 3980 as sample size since minimum number was show as 1794. Provide justification for using twice the minimum sample size.

Response 3:

Thank you for your comment. The main reason is that the attrition rate is too large in the process of questionnaire distribution control. In fact, the effect of questionnaire collection is very good. But I guess it's good to have a larger sample size, better than a sample size that can't meet the minimum sample size.

Comment 4:

Line 183: The authors  showed there are significant differences between male and female students in 3 activity groups, but we dont know where were the differences? A post hoc test would answer this question. The authors used ANOVA test but not post hoc test to show the differences between specific groups. Please include post hoc test for ANOVA tables.

Response 4:

Thank you for your comment. This is a very good suggestion. It should be noted that in the correlation analysis of physical activity variables, we quantified physical activity as high, medium, and low. Therefore, the analysis of physical activity variables is an analysis of hierarchy rather than an analysis of direct data. Therefore, the one-way ANOVA method is suitable. We very much agree with your approach to statistical analysis. Thanks again for your comment. In fact, in the paper, our analysis ideas are consistent with yours. Your idea is to determine the strength of the association between the physical activity variable and the cell phone addiction variable directly through the ANOVAs' method and post hoc test. In our analysis, ANOVAs and  post hoc test were broken down. The basic idea is this, first through one-way ANOVA to determine the difference in mobile phone addiction under different intensity physical activity. Secondly, the latent variables related to physical activity and mobile phone addiction were analyzed to determine the reasons for the differences. Finally, the linear regression method is used to determine the predictive effect between the variables. We look forward to our response to answer your questions.

Comment 5:

Line 355, the authors wrote that physical activity prevented and improved phone addiction. This statement would require an experimental study with treatment. In this case we have cross section design and correlation results. So correlation does not tell what is cause. The authors should change the narrative to be suited for obtain (correlation) results.

Response 5:

Thank you for your comment. This is a very good suggestion. We have reorganized the discussion section to be more closely aligned with the research objectives of this study.